# Effect of Vasicinone against Paraquat-Induced MAPK/p53-Mediated Apoptosis via the IGF-1R/PI3K/AKT Pathway in a Parkinson’s Disease-Associated SH-SY5Y Cell Model

**DOI:** 10.3390/nu11071655

**Published:** 2019-07-19

**Authors:** Da-Tong Ju, Kalaiselvi Sivalingam, Wei-Wen Kuo, Tsung-Jung Ho, Ruey-Lin Chang, Li-Chin Chung, Cecilia Hsuan Day, Vijaya Padma Viswanadha, Po-Hsiang Liao, Chih-Yang Huang

**Affiliations:** 1Department of Neurological Surgery, Tri-Service General Hospital, National Defense Medical Center, Taipei 114, Taiwan; 2Department of Medical Research, China Medical University Hospital, China Medical University, Taichung 404, Taiwan; 3Department of Biological Science and Technology, China Medical University, Taichung 404, Taiwan; 4Department of Chinese Medicine, Hualien Tzu Chi Hospital, Buddhist Tzu Chi Medical Foundation, Tzu Chi University, Hualien 970, Taiwan; 5School of Post-Baccalaureate Chinese Medicine, College of Chinese Medicine, China Medical University, Taichung 404, Taiwan; 6Department of Hospital and Health Care Administration, Chia Nan University of Pharmacy & Science, Tainan 717, Taiwan; 7Department of Nursing, MeiHo University, Pingtung 912, Taiwan; 8Department of Biotechnology, Bharathiar University, Tamilnadu 641046, India; 9Cardiovascular and Mitochondria related diseases research center, Hualien Tzu Chi Hospital, Hualien 970, Taiwan; 10Center of General Education, Buddhist Tzu Chi Medical Foundation, Tzu Chi University of Science and Technology, Hualien 970, Taiwan; 11Graduate Institute of Biomedicine, China Medical University and Hospital, Taichung 404, Taiwan; 12Department of Biotechnology, Asia University, Taichung 413, Taiwan

**Keywords:** parkinson’s disease, vasicinone, paraquat, apoptosis, reactive oxygen species, neuroprotection

## Abstract

Vasicinone is a quinazoline alkaloid isolated from the *Adhatoda vasica* plant. In this study, we explored the neuroprotective effect and underlying molecular mechanism of vasicinone against paraquat-induced cellular apoptosis in SH-SY5Y cells. Vasicinone reduced the paraquat-induced loss of cell viability, rescued terminal deoxynucleotide transferase-mediated dUTP nick end-labeling (TUNEL)-positive apoptotic nuclei, and suppressed generation of reactive oxygen species (ROS) in a dose-dependent manner. Western blotting analysis revealed that vasicinone increased the phosphorylation of IGF1R/PI3K/AKT cell survival signaling molecules and downregulated the paraquat-induced, mitogen-activated protein kinase (MAPK)/c-Jun N-terminal kinase (JNK)-mediated apoptotic pathways compared to that observed in cells not treated with vasicinone. This protection depended critically on the activation of IGF1R, and the silencing of IGF1R by siRNA completely abrogated the protective effect of vasicinone in SH-SY5Y cells. Our findings indicated that vasicinone is a potential candidate for the treatment of Parkinson’s disease and possibly other oxidative stress-related neurodegenerative disorders.

## 1. Introduction

Parkinson’s disease (PD) is the second most common age-dependent, chronic and progressive neurodegenerative disease and affects 2–3% of the population ≥65 years of age. It is distinguished by the selective degeneration of dopamine-producing neurons in the substantia nigra pars compacta [1], causing striatal dopamine deficiency and intracellular α-synuclein aggregation (Lewy bodies), which are the neuropathological hallmarks of Parkinson’s disease. The resulting dopamine deficiency leads to movement disorders that include rigidity, tremors at rest, slowness, involuntary movement, postural instability and freezing [2]. However, understanding the etiology and pathogenesis of PD remains elusive. Current studies have suggested that increased oxidative stress, the impairment of mitochondrial and ubiquitin-proteasome system functioning, the intensification of ions, the activation of the apoptotic cascade and loss of dopaminergic neurons play a vital role in the development of PD [3,4]. In addition, the accumulation of reactive oxygen species (ROS), mitochondrial membrane potential loss, adenosine triphosphate (ATP) depletion and the activation of the caspase cascade have been observed in the substantia nigra and cerebrospinal fluid of PD patients [5,6].

Exposure to certain environmental toxins and genetic factors can increase the risk of developing PD by affecting mitochondrial function through oxidative stress and pathological protein aggregation, contributing to neuronal cell death [7,8]. Several environmental factors, neurotoxic pollutants, some pesticides/herbicides such as 6-hydroxydopamine (6-OHDA) and 1-methyl-4-phenyl-1,2,3,6-tetrahydropyridine (MPTP), paraquat (PQ), rotenone, maneb (MB), and mancozeb (MZ) induces neurotoxicity and PD-like pathology. Paraquat (methyl viologen, 1,1′-dimethyl-4,4′-bipyridinium dichloride) is an extremely toxic herbicide widely used in agriculture to control weeds in several crops such as sorghum, soybeans, cotton, sugar cane, corn, and apple [9]. Several epidemiologic studies suggested that acute paraquat poisoning leads to severe brain damage, including cerebral edema and gliosis, and increases the incidence of PD in humans [10,11]. The chronic low-dose exposure of paraquat inducing *α*-synuclein aggregate formation, autophagy, alteration of dopamine catabolism, and inactivation of tyrosine hydroxylase causes the loss of dopaminergic cells [12]. However, based on previous reports PQ-inducing toxicity may be the best model to study dopaminergic cell death-associated PD.

A previous investigation on the role of Chinese traditional medicine in treating Parkinson’s disease revealed that approximately 22,500 medicinal herbs have anti-Parkinson’s disease activity [13]. A growing amount of evidence indicates that Chinese herbs including resveratrol, curcumin, ginsenoside, green tea polyphenols and catechins, and triptolide protect dopaminergic neurons against cell degeneration induced by the neurotoxins 1-methyl-4-phenyl-1,2,3,6-tetrahydropyridine (MPTP) or 6-hydroxydopamine (6-OHDA) [14].

The traditional medicinal plant *Adhatoda vasica*, which is associated to the Acanthaceae family, is generally known as vasaka. Its herbal extract is used to treat allergen-induced bronchial obstruction, asthma, and tuberculosis and exhibits hepatoprotective activity [15]. Previous studies in in vitro and in vivo models have reported that *Adhatoda vasica* leaf extracts have expectorant [16], bronchodilator [16,17], anti-inflammatory [18], antitussive [16] and antimicrobial activities [18] and also exhibit anticancer activities [19,20]. Vasicinone are biologically active quinazoline alkaloid found in the leaf extracts of *Adhatoda vasica*. In this study, we found that vasicinone protected SH-SY5Y cell death through the inhibition of the paraquat-induced caspase cascade via the activation of the IGF1R/PI3K/Akt signaling pathway. This is the first study to investigate whether IGF1R/PI3K/Akt-mediated signaling is accompanying in the protection of ROS-mediated apoptosis by vasicinone in SH-SY5Y cells.

## 2. Materials and Methods

### 2.1. Chemicals

Vasicinone (purity > 98%) was obtained from Cayman Chemical (CAS-486-64-6, Michigan, MI, USA). Paraquat, 2–7-diacetyl dichlorofluorescein (DCFH-DA) and 3-(4,5- dimethyl -2-thiazolyl)-2,5-diphenyltetrazolium bromide (MTT) were obtained from Sigma-Aldrich (St. Louis, MO, USA). Dulbecco’s modified Eagle medium (DMEM:F12) and fetal bovine serum (FBS) were obtained from Gibco (Grand Island, NY, USA). Primary antibodies against p-IGF1R, p-AKT, p-PI3K, Bcl-2, p-P38, p-JNK, p-ERK, p-c-Jun, p53, Bax, Bad, p-Bad, cytochrome C, and GAPDH were purchased from Santa Cruz Technology, and primary antibodies against cleaved caspase-9, cleaved caspase-3 and cleaved poly-ADP ribose polymerase (PARP) were purchased from Cell Signaling Technology.

### 2.2. Cell Culture and Treatments

Undifferentiated SH-SY5Y cells were obtained from the American Type Culture Collection (ATCC) and maintained in a Dulbecco’s modified Eagle medium/F12 nutrient mixture (DMEM:F12) with L-glutamine (Gibco, Gaithersburg, MD) supplemented with heat-inactivated fetal bovine serum (FBS) (10% v/v) and penicillin-streptomycin (1% v/v) at 37 °C under humidified atmospheric conditions containing 5% CO_2_. The cell medium was changed every 3 days.

Vasicinone was prepared in dimethylsulfoxide (DMSO), and paraquat was dissolved in phosphate buffered saline (PBS) and then stored at −20 °C. Vasicinone and paraquat were further diluted in PBS to achieve working concentrations. To examine the effect of vasicinone on paraquat-induced neurotoxicity, SH-SY5Y cells were pretreated with vasicinone (5, 7.5, 10, 15, 20 µM/mL) for 24 h and treated with paraquat (300 µM/mL) for another 24 h.

### 2.3. MTT Assay

The cell viability of SH-SY5Y cells was measured using a 3-(4,5- dimethylthiazol-2-yl)-2,5-diphenyltetrazolium bromide (MTT) assay. The cells were cultured into 96-well plates at a density of 1 × 10^4^ cells/well and then treated with different concentrations of vasicinone and paraquat for 48 h. After the treatment period, the cells were incubated with 20 µL of an MTT (5 mg/mL in PBS) solution for another 4 h at 37 °C. In DMSO the formazan crystals were solubilized and (absorbance at 570 nm) was measured using a microplate ELISA reader (Bio-Tek Instruments, Winooski, VT, USA). All experiments were accomplished independently in triplicate.

### 2.4. Annexin-V Staining Assay

SH-SY5Y cells were seeded in 6-well plates (5 × 10^5^ cells/well) for 24 h and then treated with vasicinone (10 and 15 μM) for 24 h followed by PQ (300 μM) additional 24 h. After treatment, using iced-cold PBS cells were washed and suspended in annexin V binding buffer (1 mL), which contains 5 μL of fluorochrome-labeled annexin V (FITC-V) and propidium iodide (PI). The cell suspension was incubated for 15 min at room temperature in the dark. The apoptosis rates were detected by flow cytometry (BD FACSCanto II, USA). In flow cytometry, FL1-H and FL2-H (525 and 575 nm emission filters, respectively) channels were used to detect annexin V and PI staining.

### 2.5. Measurement of Intracellular ROS Generation

The generation of intracellular ROS was measured by fluorescence with 2–7-diacetyl dichlorofluorescein (DCFH-DA). DCF-DA, a nonfluorescent probe, was deacetylated by intracellular ROS to form dichlorofluorescein (DCF). After treatment, cells were collected and incubated with DCFH-DA (5 μM/mL) at 37 °C for 30 min in the dark. Then, by using PBS the cells were washed twice, and the relative levels of DCF fluorescence were measured at a 485-nm excitation wavelength and a 535-nm emission wavelength (BD FACS Canto II, San Jose, CA, USA). Using DCF fluorescence intensity intracellular ROS levels were measured. The values were expressed as a percentage of the fluorescence compared to untreated control cells.

### 2.6. TUNEL Staining

SH-SY5Y cells were seeded into a 12-well culture plate (5 × 10^4^ cells/well) and then treated with vasicinone (10 and 15 μM/mL) for 24 h followed by PQ (300 μM/mL) for additional 24 h. After treatment, cells were washed twice with PBS, and a 4% paraformaldehyde solution in PBS was added for fixation for 1 h. Cells were washed two times with PBS, and a 3% H_2_O_2_ solution in methanol was added for 15 min for blocking. The wells were washed twice with PBS, and a permeabilization solution was added. After incubation of cells for 2 min, the wells were washed twice with PBS and incubated with the TUNEL reaction mixture (In Situ Cell Death Detection Kit, 1684817910, Roche, Mannheim, Germany) for 1 h at room temperature. After 5 min of incubation DAPI at room temperature, the cells were washed with PBS, and mounted. Photomicrographs were acquired using an Olympus DP 74 fluorescence microscope (Tokyo, Japan). The apoptosis ratio was calculated based on the number of apoptotic cells in the experimental groups divided by the total number of cells and expressed as a percentage.

### 2.7. Western Blotting Analysis

After treatment with vasicinone and PQ, SH-SY5Y cells were lysed in lysis buffer. The cell lysates were then centrifuged at 12,000× *g* for 30 min at 4 °C. The supernatant was collected for SDS-PAGE, and the total protein concentrations were measured by the Bradford method (Bio-Rad, Hercules, CA, USA). Total proteins (40 μg) were separated on polyacrylamide gels and transferred to a polyvinylidene difluoride membrane (PVDF, GE Healthcare Life Sciences, Pittsburgh, PA, USA). After protein transfer to the membrane, 5% nonfat milk in TBST was used to block the membrane at room temperature for 1 h and incubated overnight with the appropriate primary antibodies at 4 °C. After washing with TBST, the membrane was incubated with peroxidase-conjugated secondary antibody for 1 h, and the bands were visualized using enhanced chemiluminescence reagent (Millipore, Billerica, MA, USA). GAPDH was used as an internal control to ensure equal protein loading. ImageJ software (NIH, Bethesda, MD, USA) was used to do densitometric analysis and the results were normalized to their respective controls.

### 2.8. Statistical Analysis

SPSS software 17.0 (SPSS Inc., Chicago, IL, USA) used for statistical analyses. All values in the results are expressed as the mean±standard deviation (SD). Statistical analyses were performed by one-way analysis of variance (ANOVA), followed by the least significant difference post hoc test. *p* < 0.01 indicated statistical significance.

## 3. Results

### 3.1. Vasicinone Inhibited Paraquat-Induced Cytotoxicity in SH-SY5Y Cells

To evaluate the cytotoxic effects of paraquat on SH-SY5Y cells, we treated the cells to various concentrations of paraquat (100–1000 µM/mL) for 24 h (data not shown). We found that paraquat significantly reduced cell viability in a dose-dependent manner, and the cell survival rate was approximately 64% when the cells were treated with 300 µM of paraquat for 24 h compared to that observed for control cells. Thus, we used this concentration for further experiments. The different concentrations of vasicinone (1, 5, 10, 15, 20, 25, and 30 µM) had no significant effect on SH-SY5Y cell viability. The SH-SY5Y cells were subsequently pretreated with different concentrations of vasicinone (5, 7.5, 10, 15, 20 µM) for 24 h, followed by incubation with paraquat (300 µM) for another 24 h. Pretreatment with 10 and 15 µM vasicinone significantly reversed the paraquat-induced reduction in cell viability (Figure 1A).

To further evaluate the protective effect of vasicinone against paraquat-induced apoptosis, we used annexin-V/PI double staining, which was detected by flow cytometry. As shown in Figure 1B, after paraquat treatment, the total percentages of early and apoptotic cells were PQ (30.6%), PQ + 10 μM VAS (15%), PQ + 15 μM VAS (7.6%), VAS alone (6.8%) respectively. Pretreatment with vasicinone dose-dependently reduced the percentage of apoptotic cells (Figure 1B).

### 3.2. Vasicinone Prevented the Effect of Paraquat on ROS Generation in SH-SY5Y Cells

To examine paraquat-induced, ROS-mediated apoptotic cell death, we estimated intracellular ROS levels using the oxidant-sensitive probe DCFH-DA. As shown in Figure 2, paraquat treatment resulted in a significant increase in ROS generation in SH-SY5Y cells compared to that observed in untreated cells that was significantly inhibited by pretreatment with 10 μM and 15 μM vasicinone. The suppression of ROS levels by vasicinone was greater in the high-dose group than in the low-dose group (Figure 2).

### 3.3. Vasicinone Inhibited the Paraquat-Induced Activation of the JNK and p38 MAPKs in SH-SY5Y Cells

The MAPK signaling pathway is associated with progressive dysfunction in multiple neurodegenerative diseases and mitochondrial-mediated apoptosis; hence, we investigated the protective effect of vasicinone on members of the paraquat-induced MAPK pathway including JNK, ERK, c-JUN and p38 in SH-SY5Y cells. We further analyzed the phosphorylation levels of MAPK signaling molecules after treatment with vasicinone and paraquat. The protein expression levels of the p-JNK1/2, p-c-JUN and p-p38 MAPKs were increased after paraquat exposure compared to those observed without paraquat treatment, demonstrating the activation of the MAPK pathway by paraquat. Interestingly, when the cells were pretreated with vasicinone, the elevated phosphorylation of the JNK1/2, c-JUN and p38 MAPKs was inhibited, while p-ERK1/2 levels were increased compared to those observed in cells treated with paraquat alone. These results suggested that vasicinone abated the paraquat-induced injury of SH-SY5Y cells by suppressing the MAPK signaling pathway (Figure 3A,B).

### 3.4. Vasicinone Inhibited Paraquat-Induced Apoptosis in SH-SY5Y Cells

Nuclear DNA fragmentation is an important hallmark of apoptosis. To determine cell death, an in-situ terminal deoxynucleotide transferase-mediated dUTP nick end-labeling (TUNEL) assay was performed in the various treatment groups. The number of TUNEL-positive cells (green fluorescence) was dramatically higher in SH-SY5Y cells treated with paraquat than in control cells. In contrast, pretreatment with vasicinone significantly decreased these paraquat-induced changes, also there was no effect following treatment with vasicinone alone. These results confirmed that vasicinone was capable of rescuing paraquat-induced apoptotic death. This result was consistent with the decreased cell viability and increased apoptosis in paraquat-treated cells compared to those in control cells (Figure 4).

In the present study, we investigated the regulation of the p53-dependent mitochondrial intrinsic apoptotic pathway. In general, proteasome inhibition and various cellular stresses increase the levels of p53 in the cytoplasm and upregulate its downstream target, Bax, which promotes the cytochrome c release into the cytoplasm from mitochondria and activates the caspase cascade. Pretreatment with vasicinone decreased the expression of p53 induced by paraquat in SH-SY5Y cells compared to that observed in untreated cells in a dose-dependent manner (Figure 5A,B). To further investigate the protective mechanism of vasicinone against paraquat-induced apoptosis, we determined the effect of vasicinone on the expression of anti- and pro-apoptotic proteins by Western blotting. These included the Bcl-2 family proteins Bax, Bad (pro-apoptotic) and Bcl-2 (anti-apoptotic); cytochrome c, which is an essential factor of the electron transport chain; and caspase-9 and caspase-3 as inducers of the intrinsic mitochondrial apoptosis pathway. Compared to that observed in untreated cells, paraquat treatment led to the upregulation of the pro-apoptotic molecules Bax and Bad; the downregulation of anti-apoptotic Bcl-2; enhanced cytochrome c release (Figure 5A,B) and the activation of caspase-9, caspase-3 and PARP (Figure 6A,B).

In contrast, vasicinone significantly reversed the paraquat-induced activation of apoptotic signaling by enhancing Bcl-2 expression, reducing Bax expression, reducing cytochrome c release and reducing caspase and PARP cleavage. These results indicated that vasicinone restored the balance between anti-apoptotic and pro-apoptotic proteins and mitochondrial function and promoted cell survival (Figure 5 and Figure 6).

### 3.5. Vasicinone Regulated IGF-1R/PI3K/AKT and ERK Signaling Pathways in SH-SY5Y Cells

The prosurvival insulin-like growth factor 1 (IGF-1) signaling pathway can reduce oxidative stress-mediated neurodegenerative diseases. To examine the molecular mechanism underlying the protective effect of vasicinone against paraquat-induced neurotoxicity, we evaluated the IGF-1R/PI3K/AKT signaling pathway by Western blot analysis. We found that treatment with paraquat markedly decreased IGF-1R, PI3K, AKT and ERK phosphorylation, whereas pretreatment with vasicinone increased the phosphorylation of IGF-1R, AKT, PI3K and ERK compared to that observed in untreated cells. Treatment with vasicinone alone also increased the phosphorylation of IGF-1R, PI3K, AKT and ERK compared to that observed in untreated cells. These findings suggested that IGF-1 promoted cell survival and increased the phosphorylation of IGF-1R as well as its downstream targets such as AKT, PI3K and ERK1/2 in SH-SY5Y cells (Figure 7A,B).

To examine the role of IGF1R-mediated cell survival, we used siRNA-based strategy to downregulate IGF1R expression in SH-SY5Y cells. As shown in Figure 8, silencing of IGF1R by using siRNA inhibited the protein expression of p-IGF-1R, p-PI3K, p-AKT, p-ERK, and Bcl-2 and enhanced the expression of the apoptotic proteins Bax and caspase-3 compared to control cells. Therefore, this protection critically depended on the activation of IGF1R, while the knockdown of IGF1R by siRNA abrogated the expression of cell survival proteins, indicating that vasicinone protected SH-SY5Y cells via the IGF1R mediated signaling pathway (Figure 8A,B).

## 4. Discussion

In PD patients, accumulating pathological evidence has revealed an increased oxidative stress, impaired ROS/NO balance, microglia activation and chronic inflammation in the brain. Taken together, these risk factors were shown to have a detrimental effect on the integrity of dopaminergic (DA) neurons and potentially lead to neuronal cell death and subsequent neurodegeneration [21]. The etiology of PD includes both genetic and environmental factors [22,23]; however, only a small fraction of the total risk can be attributed to genetic alterations alone. Previous toxicological and epidemiological studies have demonstrated exposure pesticide is the main cause of the etiology of PD. Our results indicated that pretreatment with vasicinone attenuated apoptosis induced by paraquat, which is a commonly known neurotoxicant.

It is well known that the pathogenesis of PD is mediated through ROS generation, eventually resulting in the loss of mitochondrial membrane potential and thus leading to the activation of the caspase cascade [24]. In this study, paraquat treatment induced cellular oxidative stress, and ROS generation was the initial event mediating neuronal cell death. The present study revealed that the addition of vasicinone significantly attenuated ROS formation, which was likely responsible for the reduction of apoptosis induced by paraquat. Based on previous reports, various cellular stresses increase the level of and phosphorylate p53 in the nucleus, and p53 subsequently upregulates its target gene, Bax, which promotes the release of cytochrome c into the cytoplasm to induce apoptosis in neuronal cells [25,26,27,28]. When the cells were incubated with paraquat, the p53 level was higher than that in the control, and vasicinone treatment attenuated the p53 expression induced by paraquat in a dose-dependent manner.

The activation of the caspase cascade, DNA fragmentation, and nuclear condensation is a crucial characteristic of apoptosis. The delicate balance between Bax and Bcl-2 regulates cell integrity and controls cell survival [29] and loss of this balance activates a sequence of signaling processes leading to the induction of cell death. In mammalian cells, the caspase cascade is activated by mitochondrial cytochrome c release into the cytosol, which is regulated by Bcl-2 family proteins [30]. Our studies showed that vasicinone protected neuronal cells from paraquat-induced oxidative stress and apoptosis, as evidenced by the reduction in the ratio of Bax/Bcl-2 protein levels.

The imbalance activation of pro-apoptotic and anti-apoptotic pathways induces microglial activation followed by neuroinflammation, oxidative stress and apoptosis [31]. The activated MAP kinase signaling pathways are involved in PD pathogenesis through the release of pro-inflammatory substances by the activated microglia. These include various MAPKs such as extracellular signal-regulated protein kinase (ERK) 1/2, c-Jun NH2-terminal protein kinase (JNK), c-Jun and p38. Previous studies reported that the activation of JNK and p38 was required to inhibit the anti-apoptotic protein Bcl-2, regulating the release of cytochrome c and the induction of caspase-9 [32,33], but extracellular signal-regulated kinase (ERK) activation promoted cell survival by the anti-apoptotic signaling pathway [34]. Similar to previous findings, paraquat induced the phosphorylation of JNK, c-Jun and p38 but had no effect on ERK. Pretreatment with vasicinone had an inhibitory effect on the phosphorylation of JNK, c-Jun and p38 in a concentration-dependent manner.

Paraquat-mediated oxidative stress disturbs the balance between pro-apoptotic and prosurvival signaling. Insulin-like growth factor (IGF)-1 is a common anti-apoptotic, prosurvival factor that regulates the phosphatidylinositol-3-kinase (PI3K)/Akt downstream signaling cascade. The impaired balance of insulin-like growth factor I (IGF-I) signaling plays a significant role in the development of neurodegenerative disease [35,36]. In this study, we mainly explored the protective effect of IGF-1R/PI3K/AKT signaling against paraquat-induced cell death in SH-SY5Y cells. Our results showed that vasicinone increased the expression of IGF-1R, PI3K and AKT, which may be a major step for neuronal cell survival against paraquat toxicity. This IGF-1R activation was negated by transfection with IGF-1R siRNA in SH-SY5Y cells, which substantially inhibited the expression of p-IGF-1R, p-PI3K, p-AKT, p-ERK, and Bcl-2 and enhanced apoptotic protein expression. All these observations support vasicinone-mediated suppression of paraquat-induced cell death through the activation of IGF-1R/AKT/PI3K signaling in neuronal SH-SY5Y cells.

## 5. Conclusions

In conclusion, in this present study, we confirmed that paraquat enhanced ROS-mediated neuroinflammation, oxidative stress and apoptosis in SH-SY5Y cells. However, the effect of paraquat was counteracted by vasicinone treatment, which activated the IGF-1R/AKT/PI3K signaling pathway to inhibit MAP kinases and the expression of apoptotic proteins such as Bax and Bad; inhibited cytochrome C release; and inhibited the cleavage of caspase-9, caspase-3 and PARP, suppressing cell death. There is reasonable evidence to support vasicinone as a potential candidate for further in vivo studies.

## Figures and Tables

**Figure 1 nutrients-11-01655-f001:**
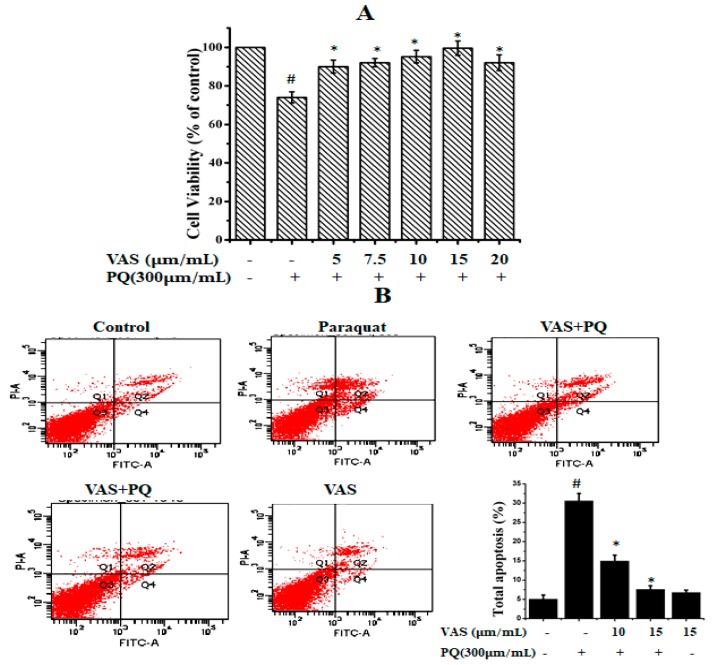
Effect of paraquat and vasicinone on SH-SY5Y cell viability. (**A**) Cells were pre-incubated with different concentrations of vasicinone for 24 h, and paraquat (300 μM) was added for an additional 24 h. Data are expressed as the percentage of the untreated control ± SD, *n* = 3. # *p* < 0.01, significantly different from control cells; * *p* < 0.01, compared with paraquat-treated group by one-way ANOVA. (**B**) Cell apoptosis and necrosis detected by flow cytometry. The results are shown as the percentage of the means and SDs for three independent experiments. # *p* < 0.01, significantly different from control cells; * *p* < 0.01, compared with paraquat-treated group. VAS: Vasicinone; VAS + PQ: Vasicinone + Paraquat.

**Figure 2 nutrients-11-01655-f002:**
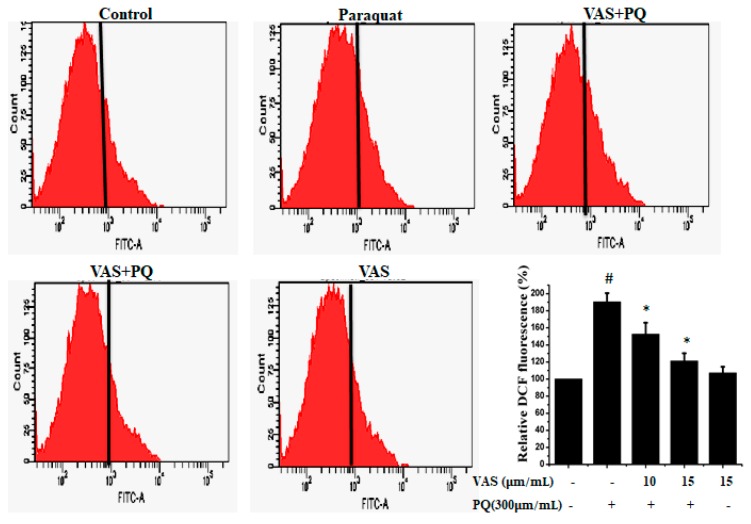
Vasicinone attenuated the paraquat-induced accumulation of reactive oxygen species (ROS) in SH-SY5Y cells. The results are expressed as the % of dichlorofluorescein (DCF) fluorescence. # *p* < 0.01, significantly different from control cells; * *p* < 0.01, compared with paraquat-treated cells. VAS: Vasicinone; VAS + PQ: Vasicinone + Paraquat.

**Figure 3 nutrients-11-01655-f003:**
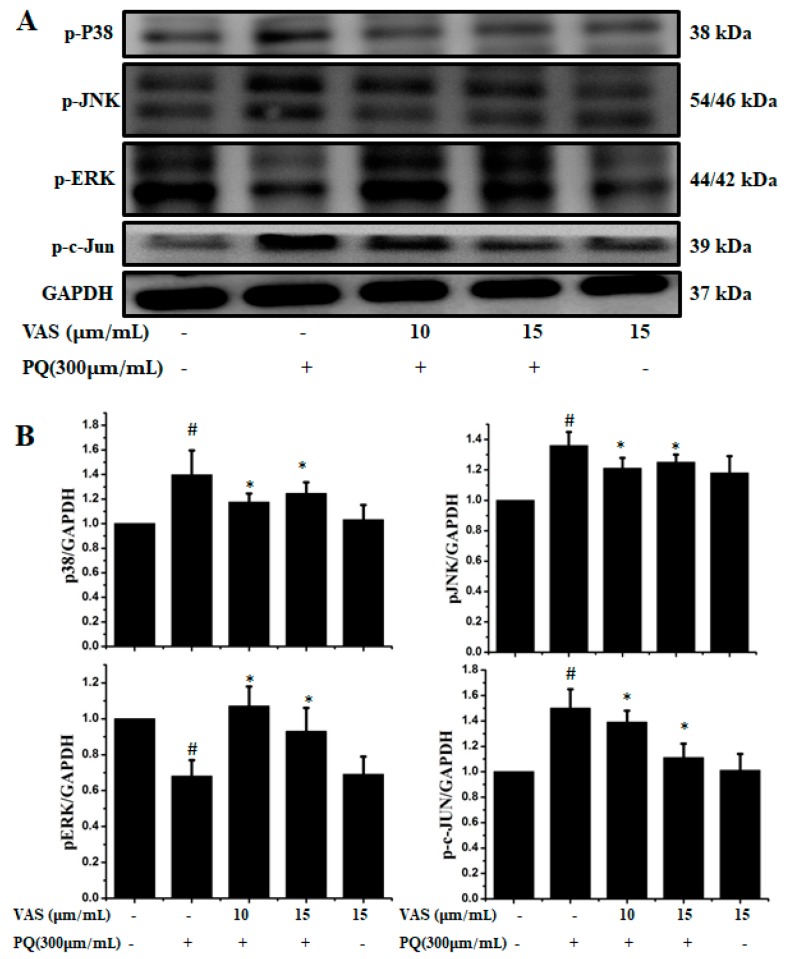
Effect of vasicinone on the paraquat-induced expression of MAP kinases in SH-SY5Y cells. (**A**) Cells were preincubated with vasicinone for 24 h, and paraquat (300 μM) was added for an additional 24 h. The protein expression levels of MAP kinases were measured using Western blotting analysis. GAPDH (glyceraldehyde-3-phosphate dehydrogenase) was used as an internal standard protein. (**B**) Phosphorylation of the p38, JNK1/2, p-ERK1/2 and c-JUN were determined by densitometry of the blots. Three independent experiments were performed. Results are shown as the mean ± SD. # *p* < 0.01, significantly different from control cells; * *p* < 0.01, compared with paraquat-treated cells.

**Figure 4 nutrients-11-01655-f004:**
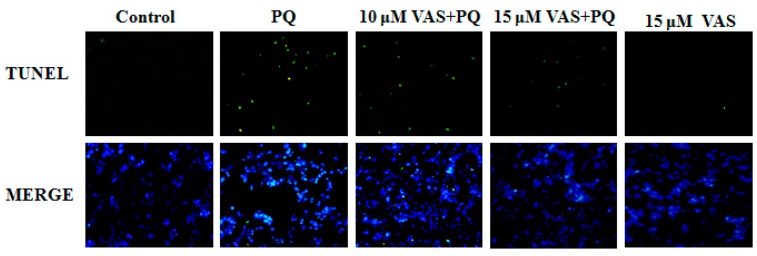
Effect of vasicinone on paraquat-induced apoptosis in SH-SY5Y cells. Terminal deoxynucleotide transferase-mediated dUTP nick end-labeling (TUNEL) staining was used to study the protective effect of vasicinone on paraquat-induced apoptosis. The green fluorescent signal was observed in TUNEL stain, and fluorescent blue (4′,6-diamidino-2-phenylindole, DAPI) signal was observed in DAPI stained nuclei. VAS: Vasicinone; VAS + PQ: Vasicinone + Paraquat.

**Figure 5 nutrients-11-01655-f005:**
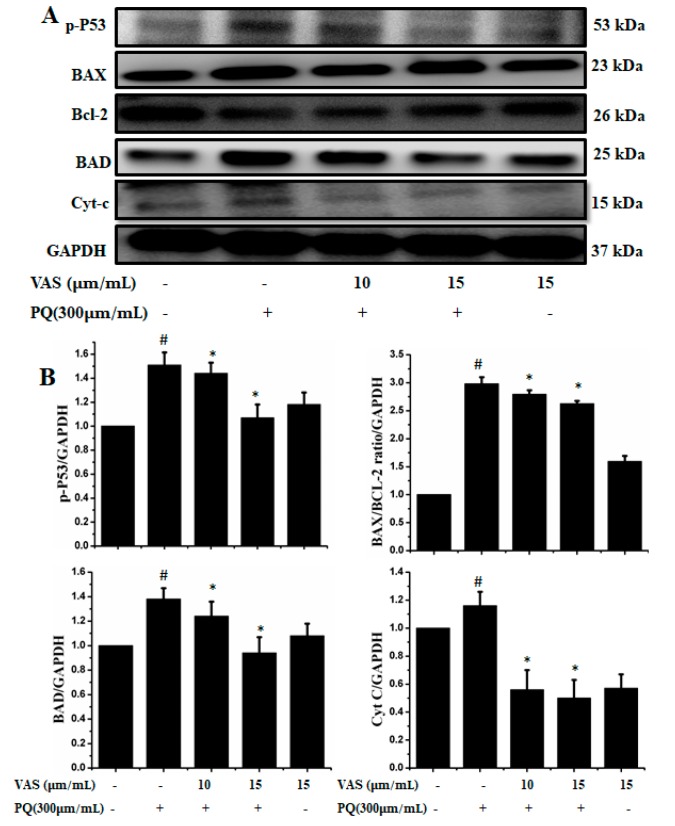
Vasicinone attenuated the paraquat-induced expression of apoptotic proteins in SH-SY5Y cells. (**A**) Cells were pretreated with vasicinone for 24 h, and paraquat was added for an additional 24 h. Total cell lysates were prepared for Western blotting analysis to determine apoptotic protein expression levels. GAPDH was used as an internal standard protein. (**B**) p53, Bax, Bad, Bcl-2 and cytochrome c release were determined by densitometry of the bands intensity. Three independent experiments were performed for this assay. Results are shown as the mean ± SD. # *p* < 0.01, compared with control cells; * *p* < 0.01, compared with paraquat-treated cells.

**Figure 6 nutrients-11-01655-f006:**
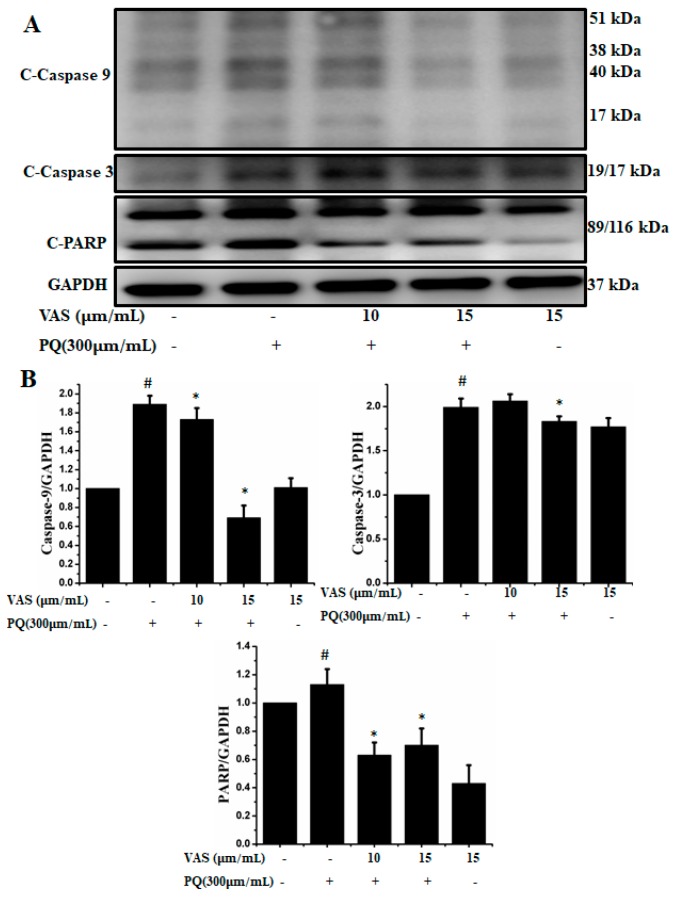
Vasicinone protected against paraquat-induced apoptosis in SH-SY5Ycells. (**A**) After pre-treatment with vasicinone for 24 h, SH-SY5Y cells were incubated with 300 μM paraquat for another 24 h. The expression level of the cleaved-caspase-9, 3 and PARP were examined by using Western blot. (**B**) Cleaved-caspase-9, 3 and PARP were determined by densitometry. Three independent experiments were performed for this assay. Results are shown as the mean ± SD. # *p* < 0.01, compared with control cells; * *p* < 0.01, compared with paraquat-treated cells.

**Figure 7 nutrients-11-01655-f007:**
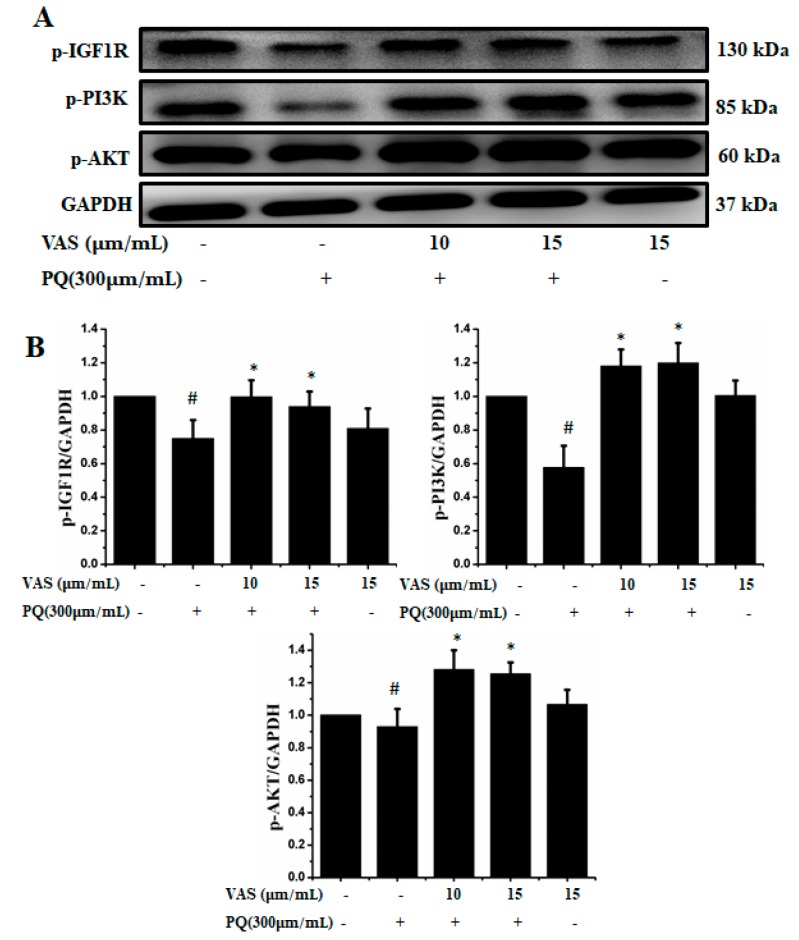
Effects of vasicinone on the activation of insulin-like growth factor 1 (IGF-1R)/PI3K/AKT signaling. (**A**) SH-SY5Y cells were treated with vasicinone for 24 h and then exposed to paraquat for an additional 24 h. Total cell lysates were prepared to determine the protein expression of cell survival-related proteins. GAPDH was used as an internal control protein. (**B**) IGF-1R, PI3K and AKT were determined by densitometry analysis. Three independent experiments were performed for this assay. Results are shown as the mean ± SD. # *p* < 0.01, compared with control cells; * *p* < 0.01, compared with paraquat-treated cells.

**Figure 8 nutrients-11-01655-f008:**
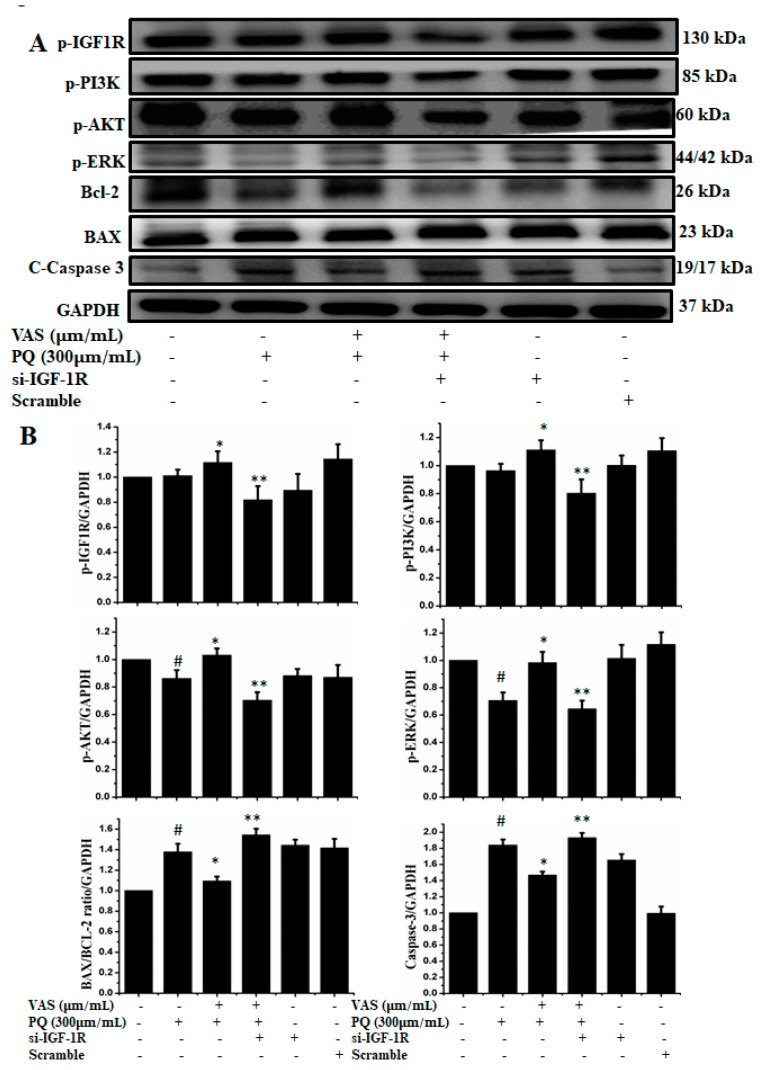
(**A**) Cells were transfected with 15 nM control siRNA (si Con) or IGF1R-targeted siRNA (si IGF1R) for 24 h, treated with 10 or 15 μM vasicinone for 24 h and then exposed to paraquat (300 μM) for an additional 24 h. GAPDH was used as an internal standard protein. (**B**) Expression of p-IGF-1R, p-PI3K, p-AKT, p-ERK, Bcl-2, apoptotic proteins Bax and caspase-3 were analyzed by densitometry of bands. Three independent experiments were performed. Results are shown as the mean ± SD. # *p* < 0.01, compared with control cells; * *p* < 0.01, compared with paraquat-treated cells; ** *p* < 0.01, compared with (Vasicinone + Paraquat) treated group.

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
