# Peer review of "Effect of Vasicinone against Paraquat-Induced MAPK/p53-Mediated Apoptosis via the IGF-1R/PI3K/AKT Pathway in a Parkinson’s Disease-Associated SH-SY5Y Cell Model"

_nutrients, 2019, doi:10.3390/nu11071655_

Reviewer 1 Report

In the paper entitled "Effect of vasicinone against paraquat-induced 2 MAPK/p53-mediated apoptosis via the IGF-3 1R/PI3K/AKT pathway in a Parkinson’s disease-4 associated SH-SY5Y cell model", by Da-Tong et al., the authors report data regarding the protective effects of Vasicinone against paraquat-induced cellular apoptosis in SH-SY5Y cells.

The authors report that the treatment with Vasicinone inhibits paraquat-induced cytotoxicity, prevents the paraquat-induced ROS generation and inhibits paraquat-induced apoptosis in SH-SY5Y cells.

Moreover, the treatment with Vasicinone regulates IGF-1R/PI3K/AKT and ERK pathways in SH-SY5Y cells.

Taken together, the reported findings are interesting and provide information on protective effects of Vasicinone, an alkaloid isolated from the Adhatoda vasica plant, that could be considered a promising candidate for the treatment of oxidative stress-related neurodegenerative disorders, as Parkinson’s disease

However, the following issues should be addressed:

- In the abstract, line 39, correct PD with Parkinson’s disease

- Do the authors have tested the effects of Vasicinon on SH-SY5Y cells viability?

- In Fig. 6, the western blot of cleaved  caspase 9 it is not of good quality, the authors should show a better one.

- The Fig. 8 is too small, the authors should show a better one.

- Page 11, lines 286-291, the sentences are confusing. Please rewrite these sentences

- Pag 13, line 355-356, the sentence “promoted the cleavage of caspase-9, caspase-3 and PARP, suppressing cell  death” is not correct, it should be “inhibit the cleavage of caspase-9, caspase-3 and PARP, suppressing cell  death”.

-The authors should check the manuscript carefully.

Reviewer 2 Report

The manuscript presented by Da-Tong Ju and colleagues shows the anti-apoptotic properties of vasicinone in vitro. In general, the manuscript is well written and organised, and presents an easy reading and understanding of the topic. The research in this work bring new insights about the anti-apoptotic pathway as a target for drug development for the treatment of PD. However, I still have several minor concerns and some major concerns that the authors should address:

Minor concerns

- Lines 54-55: The authors should rewrite the phrase in order to explain accurately the cause of PD development. From one side, we have all the causes that promote neurodegeneration of dopaminergic cells, and from the other side, we have the fact that death of dopaminergic cells is what causes all the symptoms.

- Lines 102-106: This paragraph is about the performed treatments but it does not specify the concentrations used. The authors should be more accurate.

- Lines 123-130: How did the authors measured fluorescence differences? Pixel intensity? It is not clear enough in the methods.

- Figure 1A: It looks that 300 ÎĽM PQ treatment gives a 70% cell survival. This is a high percentage. Why the authors didn't chose a higher concentration so the survival would be about 50%?

- Lines 169-171: The other concentrations used also reverse the toxicity of PQ, why the authors chose 10 and 15?

- Line 174: Could the authors specify the exact percentage for each group?

- Figure 3B: The authors compare the treatment with PQ with control, and the treatment of PQ + VAS with PQ, but what about comparing PQ + VAS with control? It looks like there are still differences in terms of expression.

- Figure 5: The same than in figure 3B

- Line 350: The authors say human neuronal SH-SY5Y cells. Did the authors differentiated the neuroblastoma cells to a neuronal phenotype? It is confusing.

Major concerns

- Line 68: The authors claim that PQ-inducing toxicity is the best model to study dopaminergic cell death-associated PD. Why PQ and not other compounds such as MPTP, MPP+, or rotenone, such as in other studies (10.3390/ijms19113369 and 10.4103/1673-5374.235250). They should include other possible models in the manuscript to give more significance to the results exposed here. Since doing the same experiments in a different model, such as for example using MPP+ instead of PQ, at least the author should mention this possibility when choosing for a model.

- Line 97: The authors should specify at least in the methods section what type of SH-SY5Y cells did they use. Where they differentiated to a dopaminergic phenotype or they were just undifferentiated? Why one or the other option?

- Figure 4: This picture is a good example of what the results describe, but is just a representative picture. Where is the raw data that allow the authors withdraw the conclusions that they state?

- Line 356-357: Claiming that this compound should be a candidate for animal and clinical studies just with the presented data is a big claim that can not be accepted. The authors should change this statement from their conclusions. It is important to highlight that due to the lack of other cellular models where these data can be compared with, the conclusions withdrawn from this study should be taken carefully. 

Author Response

Round  2

Reviewer 1 Report

The authors have provided a revised version of the paper entitled “Effect of vasicinone against paraquat-induced MAPK/p53-mediated apoptosis via the IGF-3 1R/PI3K/AKT pathway in a Parkinson’s disease-associated SH-SY5Y cell model". The authors have replied to most of the referees' objections. The manuscript is now adequately improved.

 I wish to invite the authors to give further attention to the following points:

- In Keywords: parkinson’s disease: use capital letter for Parkinson’s

- Because 300 μM PQ treatment gives a 70% cell survival, the authors should indicate in Results, lines 173-174, that “ the cell survival rate was approximately 70% when the cells were treated with 300 µM of paraquat for 24 hr compared to that observed for control cells” (not approximately 64%)

- In Results, the data regarding the cell viability after treatment with different concentration of vasicinone should be even partially included in the paper.
